# Prenatal Exposure to Parabens Affects Birth Outcomes through Maternal Glutathione S-Transferase (GST) Polymorphisms: From the Mothers and Kids Environmental Health (MAKE) Study

**DOI:** 10.3390/ijerph18063012

**Published:** 2021-03-15

**Authors:** Bohye Shin, Jeoung A. Kwon, Eun Kyo Park, Sora Kang, Seyoung Kim, Eunyoung Park, Byungmi Kim

**Affiliations:** 1Department of Population Health Sciences, Faculty of Life Sciences & Medicine, King’s College London, London WC2R 2LS, UK; bohye.shin@kcl.ac.uk; 2Institute of Health Services Research, Yonsei University, Seoul 03722, Korea; kwon.jeounga@gmail.com; 3Department of Occupational and Environmental Medicine, Ewha Medical Research Institute, Seoul 07804, Korea; ekdog1112@naver.com; 4Department of Medical Informatics, School of Medicine, Ajou University, Suwon 16499, Korea; kangsora39@gmail.com; 5Division of Cancer Prevention and Early Detection, National Cancer Control Institute, Cancer Center, Goyang 10408, Korea; seyoungkim@ncc.re.kr (S.K.); goajoa@ncc.re.kr (E.P.)

**Keywords:** birth weight, GSTM1, GSTT1, parabens, polymorphism

## Abstract

Introduction: Human exposure to parabens is very common in daily life, and prenatal exposure to these chemicals is associated with poor birth outcomes. Therefore, the aim of this study was to investigate the effect of glutathione S-transferase (GST) polymorphisms on the association between prenatal exposure to parabens and birth outcomes. Methods: We conducted a multivariate analysis involving 177 subjects to determine the association between paraben concentrations and birth outcomes in mothers with GST mu 1 (GSTM1) and GST theta 1 (GSTT1) polymorphisms from 2017 to 2019. Furthermore, we determined the interactive effect between paraben levels and GSTM1/GSTT1 polymorphisms using regression analysis, in addition to a generalized linear model after stratifying GSTM1/GSTT1 genotype into three categories. Results: Methyl and propyl paraben concentrations were significantly and positively associated with birth weight (methyl, β = 116.525, 95% confidence interval (CI) = 22.460–210.590; propyl, β = 82.352, 95% CI = 9.147–155.557) in individuals with the GSTM1-null genotype. Moreover, the propyl paraben concentration was significantly associated with an increase in gestational age (β = 0.312, 95% CI = 0.085–0.539) in individuals with the GSTM1-null genotype. Conclusions: This study reported the association between prenatal paraben exposure and birth outcomes in individuals with GST polymorphisms. We found positive relationships of maternal exposure to methyl parabens with birth weight in both mothers with GSTM1 and GSTT1-null genotypes.

## 1. Introduction

Humans are affected by various environmental factors. Environmental phenols such as parabens, triclosan, benzophenone-3, and bisphenol A are used in many industrial products. The majority of phenols are found in products used daily such as antibacterial soap, deodorant, household cleaners, kitchenware, toys, cosmetics, sunscreen, food, plastic packaging, perfumes, and plastics.

Parabens, esters of p-hydroxybenzoic acid, are one of the most encountered phenols in such products. They have chemical stability, as well as non-volatile and antibacterial properties, and thus they are used widely as preservatives in cosmetics, personal care products, food, and some medicines [1,2]. Parabens have been suspected of estrogenic activity like other endocrine-disrupting chemicals, resulting in adverse reproductive outcome [3]. It is reported that exposure to parabens can cause reproductive problems including a reduced number of sperm [4]. Moreover, maternal paraben exposure can result in increased birth weight [5]. Many recent studies have shown relationships among maternal and placental conditions, and nutrients and numerous environmental factors [6], and have stressed the importance of the influence of maternal health on birth outcome.

Birth outcomes, including weight, height, and gestational age at delivery, are significant predictors of fetal and infant health [7]. Preterm birth, defined as gestational age < 37 weeks, is a cause of neonatal death [7,8,9], and low birth weight, defined as birth weight < 2500 g, is associated with increased risks of cerebral palsy, neurological conditions, a number of developmental disabilities such as vision and hearing impairments [7], as well as asthma and chronic diseases later in life [7,8,10,11,12,13]. Both low and high birth weights are associated with negative health outcomes including death, cancer, obesity, type II diabetes mellitus, and other chronic diseases.

Several studies have demonstrated a relationship between parabens and birth outcomes. Some studies have shown a positive relationship between prenatal exposure to butyl parabens and placental weight [2,14]. The EDEN cohort study (study on the pre- and early postnatal determinants of child health and development) reported a non-significant positive association between birth weight and prenatal exposure to parabens in boys [2,15]. Moreover, other studies reported a significant positive relationship between paraben exposure and birth weight; for example, Philippat et al. [15] and Wu et al. [16] demonstrated such between methyl or propyl parabens and birth weight in boys and Aker et al. [7] between butyl parabens and birth weight. Moreover, other studies showed a relationship between maternal paraben exposure and gestational age. A positive relationship was found between methyl or propyl paraben exposure and increased gestational age, in addition to a decreased odds for small-for-gestational-age (SGA) babies [7]. Geer et al. [17] reported a negative association between gestational age and maternal urinary butyl paraben exposure during the third trimester.

In addition to environmental factors, genetics influence birth outcomes. Glutathione S-transferase (GST) enzymes, including GST theta 1 (GSTT1) and GST mu 1 (GSTM1), play significant roles in detoxification of environmental carcinogens [18,19,20,21] and in protection against genotoxic damage [22,23]. GST catalyzes the conjugation of glutathione to toxic compounds to enable their subsequent excretion [24,25,26]. Null GSTM1 and GSTT1 genotypes indicate an absence of toxin conversion [22], while other GSTM1 and GSTT1 genotypes may lead to altered individual susceptibility to environmental exposures and adverse birth outcomes [24,27,28,29,30]. The susceptibility of individuals to environmental cancers varies according to inter-individual differences in GST enzyme activity mediated by polymorphic genes [22,23].

Thus, the objective of this study was to provide evidence for an association between prenatal exposure to parabens and birth outcomes in pregnant women with GST polymorphisms, as no study has investigated this to date.

## 2. Materials and Methods

### 2.1. Population and Data Collection

This study is part of the ongoing Mothers and Kids Environmental health (MAKE) trial, which was initiated in 2017. We recruited pregnant women (≥28 weeks of gestation) who can visit study site during pregnancy in metropolitan area in Korea. A face-to-face interview was conducted to collect information, including socioeconomic status, habits of plastic and cosmetic use, and exposure to parabens. Urine and serum samples were collected at the time of visit to measure biomarkers including genetic information. We also collected the most recent sonographic findings at the time of visit and followed them up until birth to obtain data on the outcome of the birth. The study protocol was approved by the National Cancer Center of the National Cancer Center of Korea (NCC2017-0243, NCC2021-0019), and we obtained written consent from all participants prior to participation in the study.

By this time, a total of 291 pregnant women were enrolled in this study, but ultimately 243 women were visited. Six women with twin pregnancies and 2 women with no specific gravity due to insufficient urine specimens were excluded. A total of 40 women with a paraben level below the limit of detection (LOD) and 18 women with missing follow-up data on birth outcomes were excluded. Therefore, 177 participants were finally included in the analysis.

### 2.2. Urinary Concentrations of Paraben

Spot urine samples were stored in 15 mL conical tubes and refrigerated immediately at 2–6 °C. The tubes were transported within 24 h and stored in an ultra-low temperature freezer at −20 °C. High-performance liquid chromatography mass spectrometry (Agilent 6490) was used for urine separation and quantitative analysis. We used the RP18-e (100–103 mm) separator to isolate the materials for analysis, at a flow rate of 0.45 mL/min and temperature of 40 °C. Water and acetonitrile were used as the mobile phase solvents at a ratio of 85:15 (*v*/*v*), and the total analysis time was 18 min. For precise reproducibility of the data, we found that the concentrations of methyl parabens evaluated were 10, 100, and 300 µg/L (coefficient of variation (CV) 2.2–7.1%). The CV range for ethyl parabens was 3.1–6.7%, with 1, 12, and 50 µg/L, while that for propyl parabens was 3.3–5.1%, with 1, 12, and 50 µg/L. In an accuracy analysis reflecting the total error of the analysis method, the concentrations of methyl parabens evaluated were 10, 100, and 300 µg/L, representing 100 ± 5.6% values; those for ethyl parabens were 1, 12, and 50 µg/L, representing 102.3 ± 5.6% values; and those for propyl parabens were 1, 12, and 50 µg/L, representing 98.4 ± 3.8% values. The LOD was defined as the minimum concentration of parabens that could be detected in the analyte.

The standard deviation was multiplied by 3.14 (98% reliability, 6 degrees of freedom) after determining the lowest concentration of the calibration curve standard solution present in the samples. In this analysis, the LOD values were 0.116, 0.105, and 0.085 μg/L for methyl, ethyl, and propyl parabens, respectively.

Urine creatinine levels were often used to correct for dilution of urine, but creatinine levels were not stable due to rapid changes in renal creatinine clearance during pregnancy. Thus, we decided to use the urine-specific gravity, which does not depend on the factors that explain the urinary dilution. BPs concentrations were corrected for specific gravity using the formulas suggested by Cone et al. [31].
(1)$$B_{SG}=B_{m}\;\left[\frac{\left(1.018-1\right)}{\left(SG_{i}-1\right)}\right]$$
$B_{SG}$ is the specific gravity-corrected concentration (μg/L), $B_{m}$ is the measured paraben concentration, the constant 1.018 is the population median of urinary specific gravity, and $SG_{i}$ is the individual’s urine-specific gravity.

### 2.3. Genotyping Analysis

Genetic information was obtained from maternal blood samples, which were centrifuged in cryovials, divided into 1 mL aliquots, and stored at −80 °C. GST polymorphisms were verified by conducting multiplex polymerase chain reaction (PCR). Before PCR, 2 µL DNA was added to the reaction mixture, which was subsequently microcentrifuged. The PCR program consisted of 94 °C for 5 min (pre-denaturation step), 35 cycles of 94 °C for 30 s, 60 °C for 30 s, and 72 °C for 30 s, followed by 72 °C for 7 min (post-extension step) for the GSTM1 genotype, or 94 °C for 5 min (pre-denaturation step), 40 cycles at 94 °C for 40 s, 60 °C for 40 s, and 72 °C for 40 s, followed by 72 °C for 7 min (post-extension step) for the GSTT1 genotype. The amplified PCR products were subjected to agarose gel electrophoresis (2%, 250 V) and ethidium bromide staining (0.5 µg/mL) for 15 min and were observed under ultraviolet light. The GSTM1 and GSTT1 products were represented by bands at 215 and 480 bp, respectively. The internal control for GSTM1 and GSTT1 was β-globin, represented by a PCR product band at 268 bp.

### 2.4. Statistical Analysis

The general characteristics of the study subjects are expressed as numbers and percentages or as means ± standard deviation. The effects of the baseline characteristics on all birth outcomes were evaluated using Student’s *t*-test and analysis of variance. Paraben concentrations were log_10_-transformed due to their right-skewed distributions. We examined the correlation between pairs of log_10_-transformed urinary parabens concentrations using Pearson correlation coefficients. The effects of confounding factors on urinary paraben concentrations were tested using the Wilcoxon’s rank sum and Kruskal-Willis tests, and are expressed as median values due to their non-normal distributions. Concentrations below the LOD are presented as the LOD divided by the square root of 2. Confounding factors that potentially influence the effects of paraben exposure on birth outcomes were obtained from the previous literature [7,16] and included maternal age (<35 or ≥35 years), pre-pregnancy body mass index (<25.0 or ≥25.0 kg/m^2^), past history of alcohol consumption (yes or no), past history of smoking (yes or no), sex of the newborn (male or female), gestational age at delivery (<37 or ≥37 weeks), birth weight (<2500 or ≥2500 g), and parity (0 or ≥1). We decided to include patients with any missing data on birth height in the statistical analysis due to our limited sample size.

To examine interactive effect of the genotypes and paraben concentrations on birth outcomes, we included an interaction term between paraben concentrations and genotypes. After identifying the interaction effect, we performed stratified analysis by genotypes of GSTM1/GSTT1, and combined effects of GSTM1 and GSTT1 genotype were also assessed using regression analysis. To examine the effects of maternal genotype on birth outcomes, we performed regression analysis, with adjustments for covariates to prevent the effects of confounding factors. The interaction term between the genotypes and paraben levels was analyzed with relation to birth outcome.

In addition, we used a generalized linear model to stratify the GSTM1/GSTT1 genotypes into the following three categories: present, null, and double null. All statistical analyses were conducted using SAS statistical software (version 9.4; SAS Institute, Cary, NC, USA). A *p*-value < 0.05 was considered to indicate statistical significance in all statical model including interaction test.

## 3. Results

Study participant characteristics are shown in Table 1. The mean birth weight, gestational age, and birth height were 3251.32 g, 39.22 weeks, and 50.15 cm, respectively. Gestational age at birth was significantly associated with a maternal age and maternal history of alcohol consumption. The GSTM1 and GSTT1 genotypes were 54.24% and 53.11%, respectively. Moreover, the mean weight of male infants was 3341.71 g, which was approximately 200 g heavier than the mean weight of female infants at birth. The characteristics of all participants in MAKE study was shown in Appendix A.

The parabens detected at a rate > 90% are shown in Table 2. Among these, methyl parabens were detected at a rate of 100%. The geometric mean concentrations of methyl, ethyl, and propyl parabens were 36.51, 23.43, and 2.93 μg/L, respectively. Methyl paraben concentrations correlated with that in the propyl paraben concentrations during pregnancy (Pearson *r* = 0.65, *p* < 0.0001). When we analyzed the relationship between parabens according to GSTM1 and GSTT1 genotype (Figure 1), we also found that methyl and propyl parabens were strongly correlated (Pearson *r* = 0.61–0.68, *p* < 0.0001). The median concentrations of parabens stratified by the participant characteristics are shown in Table 3. Participants with a gestational age < 37 weeks had significantly higher concentrations of ethyl parabens (*p* < 0.001).

Table 4 shows the significant relationships between paraben levels and birth outcomes according to GSTM1/GSTT1 genotypes. Significant positive relationships of birth weight with methyl and propyl paraben concentrations in mothers with the null GSTM1 genotype were found. Moreover, the null GSTM1 genotype was significantly associated with gestational age and propyl paraben concentration. The relationships of the interaction between GSTM1 and methyl paraben concentrations with birth weight were significant. However, the interaction between GSTT1 and the paraben level was not significant (Appendix A).

Table 5 demonstrates the results of a combined analysis considering both the GSTM1 and GSTT1 genotypes concurrently. In both null genotypes, methyl and propyl paraben concentrations were associated with birth weight (methyl paraben: *β* = 147.77, *p* = 0.031; propyl paraben: *β* = 114.725, *p* = 0.013). No significant associations between birth outcomes and other GSTM1/GSTT1 genotypes were found.

## 4. Discussion

The MAKE study analyzed 177 women during the third trimester of pregnancy between 2017 and 2019 in Korea. The aim of this study was to investigate the effects of GSTM1 and GSTT1 polymorphisms on the relationships between maternal urine parabens, including methyl, ethyl, and propyl parabens, and birth outcomes, including birth weight, gestational age, and birth height. Other studies have analyzed the effects of GSTM1 and GSTT1 on the associations between birth outcomes and the levels of other phenols, including mercury, perfluorinated compounds, and bisphenol A. However, this is the first study to analyze the effects of GSTM1/GSTT1 on the associations between birth outcomes and parabens levels.

Our results showed the strong correlation between methyl and propyl parabens. This may be because they are the most two common parabens [32], and their mixture is mostly used in many products such as cosmetics [33] due to a synergistic effect that is more resistant to microbial contamination. On the other hand, ethyl paraben was weakly correlated with methyl and propyl parabens. This suggests that the source of ethyl paraben may be different from that of methyl and propyl parabens [34]. These findings about correlation among parabens were similar to that of the U.S. study [34].

In the analysis of the association of combined GSTM1 and GSTT1 genotypes and urinary paraben concentrations with birth outcomes, we found that methyl and propyl paraben concentrations were positively associated with birth weight in both null genotypes. Moreover, this study’s results showed a significant interaction between GSTM1 and methylparaben for their outcomes of birth weight and propylparaben for their outcomes of birth weight, gestational age, and birth height. Furthermore, there were no significant interactions with any exposures and GSTT1.

To date, several studies have investigated interactions between genetics and the environment. Chemical substances can increase the levels of polycyclic aromatic hydrocarbon DNA adducts by increasing the activity of enzymes such as GSTM1 and GSTT1 to hinder fetal and placental cellular regulation [24,35]. Oxidative stress induced by chemical exposure may increase gene–xenobiotic interactions or the production of various inflammatory cytokines in lung tissue, increasing inflammatory and immune responses. Moreover, adverse pregnancy outcomes can be induced by environmental factors and genetic GSTM1 and GSTT1 polymorphisms via modification of oxidative stress responses [24,36].

Some studies have investigated the interactions of phenol exposure and birth outcomes with GSTM1 and GSTT1-null genotypes. Kwon et al. [37] reported that GSTM1 polymorphisms may affect the negative relationship between perfluorinated compounds (PFCs) exposure and birth weight. Lee et al. [38] demonstrated a negative association between blood mercury levels and birth weight in mothers with either the GSTM1 or GSTT1-null genotype. In 2018, Lee et al. [6] reported a positive association between birth weight and bisphenol A exposure during the third trimester. Similarly, a positive association was also found between birth weight and paraben exposure during the first 3 years of life. These associations may be caused by the estrogenic action of parabens and the inability to detoxify parabens in the presence of GSTM1 and GSTT1-null genotypes [15,17]. In addition, several studies have reported an effect of phenol exposure on other birth outcome such as abdominal circumference. Ferguson et al. [14] reported that parabens were associated with a smaller abdominal circumference; however, they measured fetal abdominal circumference during pregnancy using ultrasound and did not consider GST polymorphisms. Moreover, there were several studies in adult on the relationship between chemical exposure and obesity. Hao et al. [39] described a positive association between bisphenol A levels and the incidence of central obesity in Chinese adults. Similarly, Zhang et al. [40] reported a positive relationship between phthalate levels and waist circumference in adults in the USA. Hao et al. [39] and Zhang et al. [40] measured different phenols in adults but did not take into account GST polymorphisms. Therefore, the current study is the first and important to investigate the impact of parabens on birth outcome with respect to GST polymorphisms.

Our study is not without limitations. First, the number of subjects was smaller than those in other studies that investigated the relationships between chemicals, including PFCs, mercury, and bisphenol A, and birth outcomes in mothers with GSTM1 and GSTT1-null genotypes. Second, maternal urine was used as a proxy to determine fetal paraben exposure. Third, follow-up after birth has yet to be completed. Fourth, no comparisons with other studies could be made, as this is the first study to investigate individuals with GST null genotypes in this setting. Fifth, this study used a one-time urine sample, and measurement error should be considered in interpretation. Sixth, this study had no assessment of covariates related to socio-economic status. Lastly, we evaluated the single relationship between one exposure and birth outcome. One exposure can influence a physiological effect of other exposure. Therefore, our finding needs to be confirmed using a mixture analysis considering multiple correlated exposures.

Regarding the strengths of this study, it is the first to explore the relationships between paraben levels and birth outcomes in mothers harboring GST polymorphisms. Moreover, this study analyzed each paraben (methyl, ethyl, and propyl parabens) separately and adjusted for important confounders that may have affected paraben exposure or birth outcomes. Future studies are warranted in order to determine the details of the relationships between paraben concentrations and birth outcomes in individuals with GST polymorphisms.

## 5. Conclusions

This study results showed a relationship between prenatal exposure to parabens and birth outcomes such as birth weight, gestational age, and birth height in pregnant women with GST polymorphisms. The maternal urinary concentrations of methyl and propyl parabens were positively associated with birth weight in mothers with the GSTM1 and GSTT1-null genotypes. 

## Figures and Tables

**Figure 1 ijerph-18-03012-f001:**
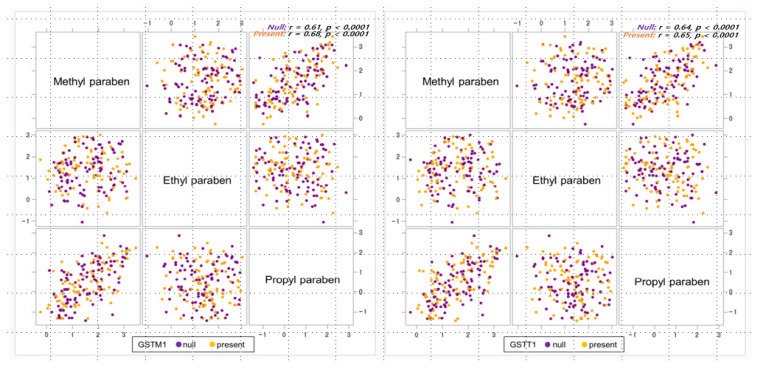
The relationship between parabens according to GSTM1 and GSTT1 genotype.

**Table 1 ijerph-18-03012-t001:** General characteristics of study subjects ^a^.

Characteristics	*n* (%)	Mean ± SD		
	Study Population	Birth Weight (g)	Gestational Age (Weeks)	Birth Height (cm)
Total	177	3251.32 ± 398.05	39.22 ± 1.50	50.15 ± 2.15
Maternal age (years)				
<35	125 (70.62)	3242.08 ± 381.45	39.37 ± 1.44 *	50.15 ± 2.04
≥35	52 (29.38)	3273.52 ± 438.52	38.86 ± 1.61	50.17 ± 2.42
Pre-pregnancy BMI (kg/m^2^)				
<25.0	152 (85.88)	3254.20 ± 383.44	39.30 ± 1.42	50.20 ± 2.04
≥25.0	25 (14.12)	3233.80 ± 486.31	38.77 ± 1.90	49.86 ± 2.75
Past history of alcohol consumption				
Yes	139 (78.53)	3257.15 ± 394.92	39.33 ± 1.32 *	50.24 ± 2.15
No	38 (21.47)	3229.97 ± 413.99	38.81 ± 1.99	49.82 ± 2.15
Past history of smoking				
Yes	159 (89.83)	3254.93 ± 377.47	39.26 ± 1.43	50.15 ± 2.07
No	18 (10.17)	3219.44 ± 561.16	38.93 ± 2.07	50.23 ± 2.80
Genotype				
GSTM1				
Present	81 (45.76)	3228.96 ± 414.77	39.20 ± 1.63	49.84 ± 2.18
Null	96 (54.24)	3277.82 ± 378.12	39.25 ± 1.35	50.53 ± 2.06
GSTT1				
Present	83 (46.89)	3248.33 ± 379.35	39.05 ± 1.65	50.19 ± 2.24
Null	94 (53.11)	3254.70 ± 420.52	39.42 ± 1.30	50.11 ± 2.05
Infant gender				
Male	90 (50.85)	3341.71 ± 396.22 ***	39.27 ± 1.40	50.53 ± 2.06
Female	87 (49.15)	3157.81 ± 379.99	39.18 ± 1.61	49.77 ± 2.18
Parity				
0	175 (74.15)	3232.10 ± 388.90	39.32 ± 1.61	50.09 ± 1.98
≥1	61 (25.85)	3293.95 ± 418.12	39.00 ± 1.21	50.30 ± 2.51

^a^*p* calculated by Student *t*-test and analysis of variance. Numbers do not always up to the same total by characteristics because of missing value. BMI, body mass index; GSTM1, glutathione S-transferase M1; GSTT1, glutathione S-transferase T1. * *p* < 0.05; *** *p* < 0.001.

**Table 2 ijerph-18-03012-t002:** Distribution of parabens concentrations (μg/L) in maternal blood.

	LOD	>LOD (%)	GM	GSD	Selected Percentiles			
					P25	P50	P75	P95
	Urinary Biomarkers (μg/L)							
Methyl paraben	0.116	100	36.51	7.01	6.35	36.37	136.76	975.68
Ethyl paraben	0.105	99.44	23.43	7.24	6.26	27.58	86.64	616.88
Propyl paraben	0.085	92.66	2.93	10.39	0.44	2.91	21.72	115.97

LOD, limit of detection; GM, geometric mean; GSD, geometric mean standard deviation.

**Table 3 ijerph-18-03012-t003:** Concentrations of parabens in maternal blood according to participant characteristics.

Characteristics	Median (μg/L)					
	Methyl Paraben	*p*-Value ^a^	Ethyl Paraben	*p*-Value ^a^	Propyl Paraben	*p*-Value ^a^
Maternal age (years)						
<35	31.10	0.82	29.56	0.36	2.33	0.93
≥35	59.28		19.07		4.33	
Pre-pregnancy BMI (kg/m^2^)						
<25	37.16	0.26	24.51	0.80	2.69	0.96
≥25.0	23.73		34.17		7.25	
Past history of alcohol consumption						
Yes	36.98	0.29	29.56	0.69	2.85	0.93
No	28.19		19.41		6.41	
Past history of smoking						
Yes	35.93	0.59	25.99	0.08	2.85	0.10
No	86.07		31.36		4.43	
Genotype						
GSTM1						
Present	35.93	0.32	29.56	0.86	3.98	0.83
Null	37.16		26.79		2.00	
GSTT1						
Present	37.92	0.63	30.63	0.74	2.19	0.89
Null	33.52		22.77		4.12	
Infant gender						
Male	31.64	0.63	39.38	0.07	2.26	0.96
Female	37.34		16.45		3.66	
Gestational age (Weeks)						
<37	36.37	0.21	32.51	0.001	24.22	0.47
≥37	36.46		27.27		2.88	
Birth weight (g)						
<2500	61.98	0.30	32.51	0.46	0.42	0.40
≥2500	36.15		27.27		2.96	
Parity						
0	31.51	0.78	25.82	0.33	2.96	0.62
≥1	39.56		31.04		2.85	

^a^*p*-Value calculated by the Wilcoxon rank sum test or Kruskal–Wallis rank sum test because of the skewness of the parabens data.

**Table 4 ijerph-18-03012-t004:** Regression coefficients and 95% confidence intervals for urinary paraben concentrations associated with birth outcomes according to GSTM1/GSTT1 genotype.

			Birth Weight (g)			Gestational Age (Weeks)		Birth Height (cm)	
Type of Paraben	Genotype		*n*	β (SE) ^a^	*p*-Value	β (SE) ^a^	*p*-Value	β (SE) ^a^	*p*-Value
Methyl paraben	GSTM1	present	81	−34.665 (34.256)	0.315	0.058 (0.137)	0.674	−0.177 (0.228)	0.441
		null	96	116.525 (47.318)	0.016	0.204 (0.152)	0.185	0.086 (0.277)	0.756
	GSTT1	present	83	53.424 (44.022)	0.229	0.159 (0.128)	0.217	−0.113 (0.244)	0.644
		null	94	43.087 (39.851)	0.283	0.181 (0.156)	0.250	0.187 (0.254)	0.465
Ethyl paraben	GSTM1	present	81	−45.753 (37.924)	0.232	0.044 (0.153)	0.772	−0.049 (0.255)	0.850
		null	96	−21.935 (44.876)	0.626	0.088 (0.141)	0.533	−0.138 (0.257)	0.593
	GSTT1	present	83	−29.006 (48.069)	0.548	0.084 (0.140)	0.548	0.049 (0.268)	0.856
		null	94	−36.788 (38.202)	0.338	0.099 (0.150)	0.510	−0.134 (0.243)	0.583
Propyl paraben	GSTM1	present	81	−11.629 (30.584)	0.705	−0.010 (0.122)	0.934	−0.233 (0.201)	0.251
		null	96	82.352 (36.825)	0.028	0.312 (0.114)	0.008	0.146 (0.212)	0.492
	GSTT1	present	83	26.954 (37.389)	0.473	0.104 (0.108)	0.341	−0.147 (0.206)	0.477
		null	94	51.013 (31.620)	0.110	0.202 (0.124)	0.107	0.171 (0.201)	0.397

^a^ All models were adjusted for mother’s age, pre-pregnancy BMI, past history of alcohol consumption, past history of smoking and gender, and parity. Birth weight and birth height models were additionally adjusted for gestational age.

**Table 5 ijerph-18-03012-t005:** Association of combined GSTM1 and GSTT1 genotypes and urinary paraben concentrations with birth outcomes.

			β (SE) ^a^					
Type of Paraben		*n*	Birth Weight (g)		Gestational Age (Weeks)		Birth Height (cm)	
			β (SE) ^a^	*p*-Value	β (SE) ^a^	*p*-Value	β (SE) ^a^	*p*-Value
Methyl paraben	Both present	50	−27.978 (52.619)	0.599	−0.012 (0.186)	0.949	−0.092 (0.269)	0.735
	Either null	90	17.613 (32.498)	0.589	0.118 (0.106)	0.268	−0.229 (0.198)	0.251
	Double null	37	147.77 (66.144)	0.031	0.29 (0.258)	0.268	0.507 (0.413)	0.227
Ethyl paraben	Both present	50	−9.617 (64.031)	0.882	−0.028 (0.225)	0.902	0.147 (0.325)	0.654
	Either null	90	−31.732 (34.937)	0.366	0.110 (0.114)	0.337	−0.013 (0.216)	0.951
	Double null	37	−31.295 (58.303)	0.594	0.042 (0.219)	0.848	−0.327 (0.346)	0.351
Propyl paraben	Both present	50	5.952 (43.619)	0.893	−0.071 (0.152)	0.647	−0.043 (0.222)	0.849
	Either null	90	4.951 (28.676)	0.863	0.073 (0.094)	0.438	−0.307 (0.174)	0.080
	Double null	37	114.725 (44.341)	0.013	0.258 (0.174)	0.147	0.469 (0.271)	0.092

^a^ All models were adjusted for mother’s age, pre-pregnancy BMI, past history of alcohol consumption, past history of smoking and gender, and parity. Birth weight and birth height models were additionally adjusted for gestational age.

## Data Availability

The data presented in this study are available on request from the corresponding author.

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
