# Peer review of "Prenatal Exposure to Parabens Affects Birth Outcomes through Maternal Glutathione S-Transferase (GST) Polymorphisms: From the Mothers and Kids Environmental Health (MAKE) Study"

_ijerph, 2021, doi:10.3390/ijerph18063012_

Round 1

Reviewer 1 Report

This study evaluates associations between prenatal paraben exposures and several birth outcomes (birth weight, gestational age and birth height) and possible modification by GST polymorphisms. The study design is generally appropriate for the questions being asked, although I think the authors should consider presenting results from interaction models rather than stratified models. Before publication, the authors should strengthen their paper by presenting results in a more interpretable form and by discussing the study results in more detail.

Introduction:

More information on GST polymorphisms, and why the authors hypothesized they would act as effect modifiers, would provide useful context for readers.

Material and Methods

Population and Data Collection

Please specify at what week of the pregnancy visits were conducted, and the range was in visit times.

Statistical Analysis

Additional information on why certain confounders were selected a priori would give the final models more credibility.

Are all models adjusted for gestational age? Presumably the model for gestational age is not. What about the fact that gestational age could be on the causal path from paraben exposure to birth weight or height?

How many patients had missing data for birth height? How big is your change in sample size?

Was the interaction term also considered significant at the 0.05 level, or was a higher cut-off used?

Results

Are the results from Table 4 from the stratified analyses? Why not show results directly from the model that includes an interaction term? When you present results from the stratified models, you add additional flexibility by allowing the effects of your other confounders (gender, parity, etc.) to vary by genotype. Showing results from the interaction model seems more reasonable, because it constrains the associations between confounders and paraben status to be the same regardless of genotype.  

Because the primary goal of this study is to evaluate modification by genotype, the results of the interaction model (Table S2) should be presented in the main paper instead of the supplementary. However, this table should be re-structured to provide a more useful interpretation of the model results. Why do the authors present the main coefficient for each paraben (presumably this is the coefficient for a null genotype, although the authors do not specify), rather than the estimated effects for genotype-null and genotype-positive, which would be more useful for readers?

Table 5: Sample size by genotype (both present, either null, double null) need to be presented. Isn’t it possible that there are significant associations between parabens and birth weight in the double-null group but not double-positive group because of differences in sample size? I am also curious why the authors chose to do a stratified analysis rather than an interaction model, which would allow them to formally test whether modification by genotype status (both present, either null, or both null) is statistically significant.

General comment on results and tables: Why don’t the authors interpret their beta coefficients in a meaningful way? If the outcomes are not transformed but the paraben exposures are log-10 transformed, the beta coefficients can be manipulated to be interpreted as the unit change in outcome for a multiplication (either a doubling, 10-fold increase, etc) in the exposure. This would be much more useful to readers.

Discussion

The authors need to discuss the results of their models in more detail. They say, “This study results showed significant interactions among maternal metabolic gene polymorphisms, maternal paraben exposure, and birth outcomes.” However, this statement is misleading and their actual results are much more nuanced. Looking at Table S2, the authors found a significant interaction between GSTM1 and both methylparaben and propylparaben for their outcomes of birth weight and birth height, but no significant associations for gestational age. Furthermore, there were no significant interactions with any exposures and GSTT1. Why do the authors think GSTM1 may act as an effect modifier, but not GSTT1? Why do they think they found associations with weight and height but not gestational age? How do these results agree or not agree with previous literature? I realize that this is the first study specifically of parabens and GST polymorphisms, but results from studies of other chemical exposures could still provide some useful context.

The authors should discuss the limitations of their study in more detail. Specifically, what are the limitations of using a one-time urine sample as a proxy for fetal paraben exposure? What is the half-life of parabens, and how representative do the authors think their sample is of fetal exposures? How could the resulting measurement error impact their study? Do fetal paraben exposures usually change over the course of a pregnancy? What have other studies found?

Do the authors believe there are other sources of unmeasured confounding in their study? This study has no assessment of covariates related to socio-economic status (income, education, employment etc). However, socio-economic status has been associated with paraben exposures, and is also an important predictor of birth outcomes.

Author Response

Response to Reviewer 1 Comments

Following are our responses to the suggestions and questions made by reviewers. We have revised the manuscript carefully according to your valuable comments and hope you may be satisfied with our revision.

This study evaluates associations between prenatal paraben exposures and several birth outcomes (birth weight, gestational age and birth height) and possible modification by GST polymorphisms. The study design is generally appropriate for the questions being asked, although I think the authors should consider presenting results from interaction models rather than stratified models. Before publication, the authors should strengthen their paper by presenting results in a more interpretable form and by discussing the study results in more detail.

Introduction:

Point 1: More information on GST polymorphisms, and why the authors hypothesized they would act as effect modifiers, would provide useful context for readers.

Response 1: Thank you for your opinion. We added more information related to GST polymorphisms in introduction below:

“GST catalyzes the conjugation of glutathione to toxic compounds to enable their subsequent excretion.(Danileviciute et al., 2012; Infante-Rivard, 2004; Raijmakers et al., 2001) Null GSTM1 and GSTT1 genotypes indicate an absence of toxin conversion,(Delpisheh et al., 2009) while other GSTM1 and GSTT1 genotypes may lead to altered individual susceptibility to environmental exposures and adverse birth outcomes.(Danileviciute et al., 2012; Hayes and Strange, 2000; Infante-Rivard et al., 2002; Infante-Rivard et al., 2006; Thier et al., 2003)”

Material and Methods. Population and Data Collection

Point 2: Please specify at what week of the pregnancy visits were conducted, and the range was in visit times.

Response 2: We appreciate your comment. We added information week of the pregnancy visits in Population and Data Collection.

Statistical Analysis

Point 3: Additional information on why certain confounders were selected a priori would give the final models more credibility.

Response 3: We appreciate the reviewer pointing out this issue. We added the references as follows: 

“Confounding factors that potentially influence the association between paraben exposure and birth outcomes were obtained from the previous literature (Aker et al., 2019; Wu et al., 2017) and finally included maternal age (< 35 or ≥ 35 years), pre-pregnancy body mass index (< 25.0 or ≥ 25.0 kg/m2), past history of alcohol consumption (yes or no), past history of smoking (yes or no), sex of the newborn (male or female), gestational age at delivery (< 37 or ≥ 37 weeks), birth weight (< 2500 or ≥ 2500 g), and parity (0 or ≥ 1).”

Point 4: Are all models adjusted for gestational age? Presumably the model for gestational age is not. What about the fact that gestational age could be on the causal path from paraben exposure to birth weight or height?

Response 4: We apologize for causing such confusion. We corrected the adjustment variables as you have checked.

Point 5: How many patients had missing data for birth height? How big is your change in sample size?

Response 5: We apologize for being unintentionally misleading. There is no missing value of birth height in analysis data. As reviewer suggested, we carefully revised the sample size of birth height in Tables.

Point 6: Was the interaction term also considered significant at the 0.05 level, or was a higher cut-off used?

Response 6: We apologize for causing such confusion. We have corrected the typos and revised the sentence about interaction p-value.

“In addition, we used a generalized linear model to stratify the GSTM1/GSTT1 genotypes into the following three categories: present, null, and double null. All statistical analyses were conducted using SAS statistical software (version 9.4; SAS Institute, Cary, NC). A p-value < 0.05 was considered to indicate statistical significance in all statical model including interaction test.”

Results

Point 7: Are the results from Table 4 from the stratified analyses?  Why not show results directly from the model that includes an interaction term? When you present results from the stratified models, you add additional flexibility by allowing the effects of your other confounders (gender, parity, etc.) to vary by genotype. Showing results from the interaction model seems more reasonable, because it constrains the associations between confounders and paraben status to be the same regardless of genotype.  Because the primary goal of this study is to evaluate modification by genotype, the results of the interaction model (Table S2) should be presented in the main paper instead of the supplementary. However, this table should be re-structured to provide a more useful interpretation of the model results. Why do the authors present the main coefficient for each paraben (presumably this is the coefficient for a null genotype, although the authors do not specify), rather than the estimated effects for genotype-null and genotype-positive, which would be more useful for readers?

Response 7: We would like to thank the reviewer for the thoughtful comment. The objective of this study was to provide evidence for an association between prenatal exposure to parabens and birth outcomes in pregnant women with GST polymorphisms. To examine interactive effect of the genotypes and paraben concentrations on birth outcomes, we included an interaction term between paraben concentrations and genotypes. After identifying the interaction effect, stratified analysis was performed by genotypes of GSTM1/GSTT1, and combined effects of GSTM1 and GSTT1 genotype were also assessed using regression analysis. As reviewer suggested, we described the stratified analysis in detail in method section.

Point 8: Table 5: Sample size by genotype (both present, either null, double null) need to be presented. Isn’t it possible that there are significant associations between parabens and birth weight in the double-null group but not double-positive group because of differences in sample size? I am also curious why the authors chose to do a stratified analysis rather than an interaction model, which would allow them to formally test whether modification by genotype status (both present, either null, or both null) is statistically significant.

Response 8: We appreciate the reviewer pointing out this issue. As reviewer suggested, we provided sample size in Table5 and described the stratified analysis in detail in method section.

Point 9: General comment on results and tables: Why don’t the authors interpret their beta coefficients in a meaningful way? If the outcomes are not transformed but the paraben exposures are log-10 transformed, the beta coefficients can be manipulated to be interpreted as the unit change in outcome for a multiplication (either a doubling, 10-fold increase, etc) in the exposure. This would be much more useful to readers.

Response 9: We would like to thank the reviewer for the thoughtful comment. This study had some limitations. The sample size of this study was not enough to obtain statistical power. Therefore, our findings should be confirmed using in large scale-prospective studies. We are very cautious about giving the effect size. We hope you focus on the significance of the relation rather than the effect size.

Discussion

Point 12: The authors need to discuss the results of their models in more detail. They say, “This study results showed significant interactions among maternal metabolic gene polymorphisms, maternal paraben exposure, and birth outcomes.” However, this statement is misleading and their actual results are much more nuanced. Looking at Table S2, the authors found a significant interaction between GSTM1 and both methylparaben and propylparaben for their outcomes of birth weight and birth height, but no significant associations for gestational age. Furthermore, there were no significant interactions with any exposures and GSTT1. Why do the authors think GSTM1 may act as an effect modifier, but not GSTT1? Why do they think they found associations with weight and height but not gestational age? How do these results agree or not agree with previous literature? I realize that this is the first study specifically of parabens and GST polymorphisms, but results from studies of other chemical exposures could still provide some useful context.

Response 12: Thank you for your comments. As you mentioned, we modified sentences. 

This study results showed significant interactions among maternal metabolic gene polymorphisms, maternal paraben exposure, and birth outcomes. Also, this study results showed a significant interaction between GSTM1 and methylparaben for their outcomes of birth weight and propylparaben for their outcomes of birth weight, gestational age, and birth height. Furthermore, there were no significant interactions with any exposures and GSTT1.”

Point 13: The authors should discuss the limitations of their study in more detail. Specifically, what are the limitations of using a one-time urine sample as a proxy for fetal paraben exposure? What is the half-life of parabens, and how representative do the authors think their sample is of fetal exposures? How could the resulting measurement error impact their study? Do fetal paraben exposures usually change over the course of a pregnancy? What have other studies found? Do the authors believe there are other sources of unmeasured confounding in their study? This study has no assessment of covariates related to socio-economic status (income, education, employment etc). However, socio-economic status has been associated with paraben exposures, and is also an important predictor of birth outcomes.

Response 13: Thank you for your comments. As you mentioned, we added more contents in limitations.

“Forth, this study used a one-time urine sample and measurement error should be considered in interpretation. Fifth, this study has no assessment of covariates related to socio-economic status.”

Reviewer 2 Report

In the article "Prenatal exposure to parabens affects birth outcomes through maternal Glutathione S-Transferase (GST) Polymorphisms: from the Mothers and Kids Environmental health (MAKE) Study" the authors have done a good job.  the article is well written and easy to understand. 

A minor spell check is required though.

It is better to put Pearson correlation and p-value in figure1 as well.  

Author Response

Response to Reviewer 2 Comments

Following are our responses to the suggestions and questions made by reviewers. We have revised the manuscript carefully according to your valuable comments and hope you may be satisfied with our revision.

Point 1: In the article "Prenatal exposure to parabens affects birth outcomes through maternal Glutathione S-Transferase (GST) Polymorphisms: from the Mothers and Kids Environmental health (MAKE) Study" the authors have done a good job.  the article is well written and easy to understand.  A minor spell check is required though.

It is better to put Pearson correlation and p-value in figure1 as well.

Response 1: Thank you for your comment. We checked spells as you mentioned. We also have added Pearson's correlation coefficient r with p-value in figure1.

Reviewer 3 Report

The authors tested the association between paraben concentrations and birth outcomes in mothers with glutathione S-transferases (GST) polymorphisms. The authors conducted a multivariate analysis involving 177 individuals. The authors found a positive correlation of maternal exposure to methyl parabens with birth weight in mothers with GST M1 and GST T1 null genotypes. The study is important in its field. The reviewer recommends acceptance after corrections.

Comments:

Authors should explain why measured methyl and propyl paraben levels correlate during pregnancy, while ethyl paraben was weakly correlated with methyl and propyl parabens, 

Could these linked variables influence the significant findings they report for GST null genotypes? For example, if ethyl paraben was merely correlated to propyl paraben, it would not have a physiological relationship to the outcome.

Authors should explain in greater detail in the introduction, what parabens are and what their chemical properties are, as the introduction seems a bit superficial.

The three last paragraphs of the paper are stating the same thing, too redundant.

Paragraph on GST should go to the introduction, not discussion:

"Several important enzymes influence detoxification, including GST. GST catalyzes the conjugation of glutathione to toxic compounds to enable their subsequent excretion. Null GSTM1 and GSTT1 genotypes indicate an absence of toxin conversion, while other GSTM1 and GSTT1 genotypes may lead to altered individual susceptibility to environmental exposures and adverse birth outcomes."

The poor English language needs to be corrected, e.g.:

"This study was to provide evidence for an relationship"

Author Response

Response to Reviewer 3 Comments

Following are our responses to the suggestions and questions made by reviewers. We have revised the manuscript carefully according to your valuable comments and hope you may be satisfied with our revision.

The authors tested the association between paraben concentrations and birth outcomes in mothers with glutathione S-transferases (GST) polymorphisms. The authors conducted a multivariate analysis involving 177 individuals. The authors found a positive correlation of maternal exposure to methyl parabens with birth weight in mothers with GSTM1 and GST T1 null genotypes. The study is important in its field. The reviewer recommends acceptance after corrections.

Comments:

Point 1: Authors should explain why measured methyl and propyl paraben levels correlate during pregnancy, while ethyl paraben was weakly correlated with methyl and propyl parabens.

Response 1: We agree. As reviewer suggested, we added the explanation about exposure to toluene from work in Discussion section.

“Our results showed the strong correlation between methyl and propyl parabens. This may be because they are the most two common parabens32 and their mixture are most used in many products such as cosmetics33 due to a synergistic effect that are more resistant to microbial contamination. On the other hand, ethyl paraben was weakly correlated with methyl and propyl parabens. These suggest that the source of ethyl paraben may be different from that of methyl and propyl parabens.34 These findings about correlation among parabens were similar to that of the US study.34

Point 2: Could these linked variables influence the significant findings they report for GST null genotypes? For example, if ethyl paraben was merely correlated to propyl paraben, it would not have a physiological relationship to the outcome.

Response 2: We agree. We think that further research is needed to supplement this part. As such, we reflected it as a limitation.

 “Lastly, we evaluated the single relationship between one exposure and birth outcome. One exposure can influence a physiological effect of other exposure. Therefore, our finding needs to be confirmed using a mixture analysis that consider multiple correlated exposures”

Point 3: Authors should explain in greater detail in the introduction, what parabens are and what their chemical properties are, as the introduction seems a bit superficial.

Response 3:  We agree. As reviewer suggested, we added it in the introduction as below.

“Parabens, esters of p-hydroxybenzoic acid, are one of the most encountered phenols in daily products. They have chemical stability, non-volatile, and antibacterial properties, so they are used widely as preservatives in cosmetics, personal care products, food, and some medicines 1,2. Parabens have been suspected of estrogenic activity like other endocrine-disrupting chemicals, resulting in adverse reproductive outcome.3 It is reported that exposure of paraben can cause reproductive problems including a reduced number of sperm.4 Moreover, maternal paraben exposure can result in increased birth weight.5 any recent studies have shown relationships among maternal and placental conditions and nutrients and numerous environmental factors,6 and have stressed the importance of the influence of maternal health on birth outcome.”

Point 4: The three last paragraphs of the paper are stating the same thing, too redundant.

Response 4: Thank you for your opinion. As you mentioned, we modified redundant paragraphs.

“Regarding the strengths of this study, it is the first to explore the relationships between paraben levels and birth outcomes in mothers harboring GST polymorphisms. Also, this study analyzed each paraben (methyl, ethyl, and propyl parabens) separately and adjusted for important confounders that may have affected paraben exposure or birth outcomes. Future studies are warranted to determine the details of the relationships between paraben concentrations and birth outcomes in individuals with GST polymorphisms.

  1. Conclusion

This study results showed a relationship between prenatal exposure to parabens and birth outcomes such as birth weight, gestational age, and birth height in pregnant women with GST polymorphisms. The maternal urinary concentrations of methyl and propyl parabens were positively associated with birth weight in mothers with the GSTM1 and GSTT1 null genotypes.”

Point 5: Paragraph on GST should go to the introduction, not discussion:

Response 5: Thank you for your comment. We moved this paragraph to the introduction.

"Several important enzymes influence detoxification, including GST. GST catalyzes the conjugation of glutathione to toxic compounds to enable their subsequent excretion. Null GSTM1 and GSTT1 genotypes indicate an absence of toxin conversion, while other GSTM1 and GSTT1 genotypes may lead to altered individual susceptibility to environmental exposures and adverse birth outcomes."

Point 6: The poor English language needs to be corrected, e.g.: "This study was to provide evidence for an relationship"

Response 6: Thank you for your opinion. We modified this sentence.

“This study results showed a relationship between prenatal exposure to parabens and birth outcomes such as birth weight, gestational age, and birth height in pregnant women with GST polymorphisms.”

Reviewer 4 Report

The authors analyzed the relationship between urinary parabens levels and birth outcomes in 177 mothers with GST polymorphisms. Their results claimed that the methyl and propyl parabens concentration in urine was positively corrected with birth weight in women with GSTM1 and GSTT1 null genotypes.

I have some questions:

  1. The GST polymorphisms play the main roles in this study. However, the authors did not provide a clear background to introduce how these polymorphisms affect GST enzyme activity and how they affect parabens' metabolism.
  2. Fig.1 is an important result to show the relationship between parabens and GST genotypes. I did not understand this figure; what is the scale?
  3. What is the meaning of the correlation between GST polymorphisms with paraben concentration and with birth outcome in the view of public health?
  4. The last sentence in method 2.4 statistical analysis: p-value should be < 0.05.

Author Response

Response to Reviewer 4 Comments

Following are our responses to the suggestions and questions made by reviewers. We have revised the manuscript carefully according to your valuable comments and hope you may be satisfied with our revision.

The authors analyzed the relationship between urinary parabens levels and birth outcomes in 177 mothers with GST polymorphisms. Their results claimed that the methyl and propyl parabens concentration in urine was positively corrected with birth weight in women with GSTM1 and GSTT1 null genotypes.

I have some questions:

Point 1: The GST polymorphisms play the main roles in this study. However, the authors did not provide a clear background to introduce how these polymorphisms affect GST enzyme activity and how they affect parabens' metabolism.

Response 1: Thank you for your comment on background. We added more information related to GST polymorphisms, environmental exposures, and adverse birth outcomes in introduction.

“GST catalyzes the conjugation of glutathione to toxic compounds to enable their subsequent excret.(Danileviciute et al., 2012; Infante-Rivard, 2004; Raijmakers et al., 2001) Null GSTM1 and GSTT1 genotypes indicate an absence of toxin conversion,(Delpisheh et al., 2009) while other GSTM1 and GSTT1 genotypes may lead to altered individual susceptibility to environmental exposures and adverse birth outcomes.(Danileviciute et al., 2012; Hayes and Strange, 2000; Infante-Rivard et al., 2002; Infante-Rivard et al., 2006; Thier et al., 2003)

Point 2: Fig.1 is an important result to show the relationship between parabens and GST genotypes. I did not understand this figure; what is the scale?

Response 2: Thank you for your comment. We have added Pearson's correlation coefficient r with p-value in figure1.

Point 3: What is the meaning of the correlation between GST polymorphisms with paraben concentration and with birth outcome in the view of public health?

Response 3: Thank you for your comment on the view of public health. Prior to this study, if public health was treated with a limited perspective on exposure to chemicals and birth outcomes, through this study, genetics and environmental factors such as GST polymorphism were simultaneously considered to find out the effects on birth outcomes. Also, it is possible to develop preventive medicine.

Point 4: The last sentence in method 2.4 statistical analysis: p-value should be < 0.05.

Response 4: Thank you for your comment. We fixed p-value.

“A p-value <0.05 was considered to indicate statistical significance.”